# Integrated Microbiota and Metabolome Analysis to Assess the Effects of the Solid-State Fermentation of Corn–Soybean Meal Feed Using Compound Strains

**DOI:** 10.3390/microorganisms11051319

**Published:** 2023-05-17

**Authors:** Yue Li, Qinghong Hao, Chunhui Duan, Yawei Ding, Yuanyuan Wang, Xiaojun Guo, Yueqin Liu, Yunxia Guo, Yingjie Zhang

**Affiliations:** 1College of Life Sciences, Hebei Agricultural University, Baoding 071001, China; 18832825532@163.com (Y.L.); hqh1128@163.com (Q.H.); 17731760957@163.com (Y.W.); 2College of Animal Science and Technology, Hebei Agricultural University, Lokai South Street, Baoding 071001, China; duanchh211@126.com (C.D.); dingyawei999@163.com (Y.D.); liuyueqin66@126.com (Y.L.); 3Hebei Province Feed Microorganism Technology Innovation Center, Baoding 071001, China; guoxiaojun545@126.com

**Keywords:** microbial communities, feed fermentation, untargeted metabolomics, *Bacillus licheniformis*, *Bacillus subtilis*, lactic acid bacteria, 16S rDNA sequencing

## Abstract

Solid-state fermentation is known to improve plant-based feed nutritional quality; however, the association between microbes and metabolite production in fermented feed remains unclear. We inoculated corn–soybean–wheat bran (CSW) meal feed with *Bacillus licheniformis* Y5-39, *Bacillus subtilis* B-1, and lactic acid bacteria RSG-1. Then, 16S rDNA sequencing and untargeted metabolomic profiling were applied to investigate changes in the microflora and metabolites, respectively, and their integrated correlations during fermentation were assessed. The results indicated that trichloroacetic acid soluble protein levels showed a sharp increase, while glycinin and β-conglycinin levels showed a sharp decrease in the fermented feed, as confirmed by sodium dodecyl sulfate–polyacrylamide gel electrophoresis. *Pediococcus*, *Enterococcus*, and *Lactobacillus* were predominant in the fermented feed. Overall, 699 significantly different metabolites were identified before and after fermentation. Arginine and proline, cysteine and methionine, and phenylalanine and tryptophan metabolism were the key pathways, with arginine and proline metabolism being the most important pathway in the fermentation process. By analyzing the correlation between the microbiota and metabolite production, lysyl–valine and lysyl–proline levels were found to be positively correlated with *Enterococcus* and *Lactobacillus* abundance. However, *Pediococcus* was positively correlated with some metabolites contributing to nutritional status and immune function. According to our data, *Pediococcus*, *Enterococcus*, and *Lactobacillus* mainly participate in protein degradation, amino acid metabolism, and lactic acid production in fermented feed. Our results provide new insights into the dynamic changes in metabolism that occurred during the solid-state fermentation of corn–soybean meal feed using compound strains and should facilitate the optimization of fermentation production efficiency and feed quality.

## 1. Introduction

Corn–soybean–wheat bran (CSW) meal feed, which is widely used for animal feed, contains diverse antinutritional factors, such as soybean antigenic proteins (glycinin and β-conglycinin), soybean trypsin inhibitor, and oligosaccharides. These are known to interfere with the bioavailability of nutrients in animals, specifically young animals [1]. Soybean antigenic proteins in the diets of young animals provoke a transient hypersensitivity related to an abnormal morphology of the small intestine; these changes induce malabsorption syndrome, growth depression, and diarrhea [2]. Fermentation is an effective method of reducing antinutritional factor levels and increasing feed nutritional value and digestibility [3,4,5]. The nutritional characteristics of fermented feed are predominantly related to microbial abundance. Fermenting microbes, such as *Bacillus subtilis* and *Enterococcus faecalis*, can degrade glycinin and β-conglycinin in soybean and consequently improve corn–SBM feed peptides and amino acids. Wang et al. (2022) showed that phenylalanine metabolism is the most important pathway in the fermentation of corn–soybean by *Bacillus subtilis* [6]. *Lactic acid* bacteria produce organic acids, improve feed palatability, and lengthen feed shelf-life [7]. Zhao et al. (2021) [8] reported that during fermentation, levels of saccharides, amino acids, and nucleosides decreased, while bioactive molecules in barley were released and metabolites were accumulated by *Lactobacillus plantarum* dy-1. It has been widely reported that fermented feed can improve the growth and immune function of young animals and decrease the incidence of diarrhea [9,10,11].

Microbes and their metabolites in fermented feed majorly determine the quality of the final feed [6,12]. At present, high-throughput sequencing and “omics” technologies represent reliable tools for comprehensive analyses of microbial communities and for determining the metabolite profile of fermented feed [13,14]. Wang et al. (2022) [6] inoculated *B. subtilis* and *E. faecalis* into corn and defatted soybean to achieve a two-stage solid-state fermentation (SSF) and performed 16S sequencing and liquid chromatography (LC)–tandem mass spectrometry (MS/MS) to assess the dynamics of the microbiota, metabolites, and their integrated correlations during fermentation; different metabolites were identified before and after fermentation, and protein degradation, amino acid synthesis, and carbohydrate metabolism were the main metabolic pathways. In SSF, the metabolic actions of diverse microorganisms, including fungi and bacteria, evidently play a key role in determining the quality of the final feed [15]. However, the influencing mechanism of composite microbial fermentation on the special quality of CSW feed remains unclear. Li et al. (2018) [16] performed shotgun metagenomic and metabolomic analyses to report significant variations in the composition of microbiota, collective functional genes, and flavor compounds during SSF. Such methods seem to be effective to decipher the influence of the total microbial community and microbial secondary metabolites on nutrient composition and the bioavailability of fermented feed as they can highlight not only the structure of total microbial communities but also the metabolic potential and functional profiles of microbial communities. Considering that fermenting microbes are a key impacting factor, in this study we inoculated *B. licheniformis* Y5-39, *B. subtilis* B-1, and RSG-1 into CSW feed and performed SSF. Further, 16S rDNA sequencing and untargeted metabolomic profiling were conducted to determine changes in the microflora and metabolites, respectively, and to assess their integrated correlations during fermentation. Our findings should enhance our understanding of how to improve the quality of fermented CSW feed.

## 2. Material and Methods

### 2.1. Experimental Design and Sampling

*B. subtilis* B-1 (CGMCC no. 22064), *B. licheniformis* Y5-39 (CGMCC no. 22062) [17], and RSG-1 (CGMCC no. 22061) were isolated from the intestine of healthy sheep for *Escherichia coli* growth inhibition by the College of Life Sciences, Hebei Agricultural University (Baoding, China) and stored at the China General Microbiological Culture Collection Center (CGMCC, Beijing, China).

*B. subtilis* B-1 and *B. licheniformis* Y5-39 were cultured in nutrient liquid media (0.3% beef extract, 0.5% NaCl, 1% peptone, pH 7.2–7.3) at 37 °C for 24 h and lactic acid bacteria RSG-1 was cultured in de Man, Rogosa, and Sharpe broth at 37 °C for 18 h prior to fermentation. To achieve the best fermentation performance, *B. subtilis* B-1 (1.0 × 10^8^ CFU/mL), *B. licheniformis* Y5-39 (1.0 × 10^8^ CFU/mL), and lactic acid bacteria RSG-1 (1.0 × 10^8^ CFU/mL) were mixed in a volume ratio of 1:4:1 [18]. Approximately 150 g of the substrate comprised 60% corn, 25% soybean, and 15% wheat bran. The fermented group was inoculated with 10% compound microbial agents, and sterile water was added to achieve a 45% moisture content. The unfermented group was not inoculated with compound microbial agents. All procedures remained the same for the unfermented and fermented feed, except that sterile media (nutrient broth/de Man, Rogosa, and Sharpe broth) were added to the unfermented feed instead of bacteria. The wet mixture was then transferred to a plastic fermentation bottle (200 mL) and fermented at 37 °C for 14 days.

### 2.2. Measurement of Nutritional Content

Moist fermented and unfermented feed (n = 6) were collected on day 14 to determine the count of microorganisms and levels of microbial metabolites, and the remaining samples were dried at 65 °C for 48 h, crushed, and subjected to physicochemical analyses. Dried samples were used to analyze crude protein (CP) levels, as recommended by the AOAC International guidelines (2005). Trichloroacetic acid soluble protein (TCA-SP) levels in the fermented and unfermented feed were measured as previously described by Ovissipour et al. (2009) [19], and glycinin and β-conglycinin levels in the fermented and unfermented feed were measured using an indirect competitive ELISA kit (Longkefangzhou Bio-Engineering Technology Co., Ltd., Beijing, China). The pH was measured using a pH meter (PSH-3D, Mettler Toledo, Switzerland) as described by Shi et al. (2017) [20]. Lactic acid content was determined using a lactic acid enzymology assay kit (Nanjing Jiancheng Bio Co., Nanjing, China) as per the manufacturer’s instructions.

### 2.3. Sodium Dodecyl Sulfate–Polyacrylamide Gel Electrophoresis (SDS-PAGE) and LC–MS/MS

Soluble proteins were extracted from the unfermented and fermented feed samples as previously described [21,22], with minor modifications. Protein extracts (0.05 g/mL) were dissolved in Tris-HCl buffer (1.5 M, pH 8.8), mixed with a loading buffer (4:1, *v/v*), and then heated for 10 min in boiling water. A 10-μL aliquot of this mixture was then loaded onto an SDS-PAGE gel (12% separating gel and 5% stacking gel, Bis-Tris). The operating voltage of the stacking and separating gels was 80 V and 133 V, respectively. A protein ladder (10–180 kDa) was used as a size marker. After electrophoresis, gels were stained with Coomassie Brilliant Blue R-250 for 40 min and de-stained overnight in 7% acetic acid.

Protein bands (<25 kDa) were excised. LC–MS/MS was performed to parse soybean proteins and peptides. Briefly, after in-gel digestion by trypsin, 5 μL of the total peptides were separated and analyzed with a nano-UPLC (EASY-nLC1200) coupled to a Q Exactive HFX Orbitrap instrument with a nano-electrospray ion source.

### 2.4. 16S rDNA Sequencing

Microbial DNA was extracted from the fermented and unfermented feed samples using the HiPure soil DNA kit (Magen, Guangzhou, China), as per the manufacturer’s instructions. DNA concentration and purity were determined by NanoDrop 2000 (Thermo Fisher Scientific, Waltham, MA, USA). The universal primers 338F (5′-ACTCCTACGGGAGGCAGCA-3′) and 806R (5′-GGACTACNNGGGTATCTAAT-3′) were used to amplify the V3–V4 region of the bacterial 16S rDNA gene. Both the forward and reverse 16S primers were tailed with sample-specific Illumina index sequences to allow for deep sequencing. A Phusion^®^ High-Fidelity PCR Master Mix and a GC Buffer kit (New England Biolabs, Ipswich, MA, USA) were used to amplify the bacterial 16S rDNA gene by PCR (cycling conditions: 95 °C for 5 min, followed by 30 cycles at 95 °C for 1 min, 60 °C for 1 min, 72 °C for 1 min, and a final extension at 72 °C for 7 min). Subsequently, amplicons were subjected to electrophoresis on 2% agarose gels, purified with Agencourt AMPure XP Beads (Beckman Coulter, Bria, CA, USA), and then finally quantified using the Qubit dsDNA HS assay kit and Qubit 4.0 Fluorometer (Invitrogen, Thermo Fisher Scientific, Waltham, MA, USA). Purified amplicons were pooled in equimolar ratios and paired-end sequenced (PE250) on Illumina Novaseq 6000 as per standard protocols. Raw reads can be obtained by contacting the corresponding authors.

High-quality clean reads were obtained as follows: raw reads were processed by Trimmomatic v0.33, and clean reads were obtained after primer sequences were identified and removed by Cutadapt 1.9.1. Chimeric sequences were detected and removed with UCHIME v4.2 to obtain clean reads. Overall, 478,508 clean reads were obtained, with >79,751 clean reads for each sample. Filtered sequences were clustered into operational taxonomic units of ≥97% similarity using the USEARCH pipeline [23]. Representative sequences were then categorized into organisms via a naive Bayesian model with an RDP classifier on the basis of the SILVA v132 database [24,25]. Principal coordinate analysis (PCoA) was performed based on the Bray–Curtis distance metric [26], and to estimate α-diversity, the number of observed species and the indices of Chao1 (species richness) and Shannon and Simpson (diversity) were calculated. Linear discriminant analysis (LDA) (threshold > 4.0) and LDA effect size (LEfSe) analysis [27] were applied to identify species with significant differences in microbial succession after CSW feed fermentation. PICRUSt2 [28] was used to predict the function of composition of samples and to speculate on functional gene composition and differences between the groups.

### 2.5. Untargeted Metabolomics

A 50 mg feed sample was thawed on ice and metabolites were extracted with 1000 μL of extract solution (methanol: acetonitrile: water = 2:2:1, with 20% 2-chloro-L-phenylalanine as the internal standard), as previously reported [29]. Quality control samples were prepared by mixing equal aliquots of supernatants from all samples. LC–MS/MS was performed on an UHPLC–MS/MS system (UPLC, Waters Acquity I-Class PLUS, Milford, MA, USA) involving a high-resolution mass spectrometer (Waters Xevo G2-XS QTOF, Milford, MA, USA) and column (1.8 μm, 2.1 × 100 mm, Waters Acquity UPLC HSS T3, Milford, MA, USA). Quadrupole time-of-flight mass spectrometer was operated in both positive and negative ion modes. MassLynx v4.2 was used to collect high-resolution MS data, which were converted to the mzXML format. Subsequently, data were imported into Progenesis QI for peak extraction, peak alignment, and other data processing operations based on the Progenesis QI online METLIN database and Biomark’s self-built library for identification; at the same time, theoretical fragment identification was performed [30].

After normalizing the original peak area information with the total peak area, multivariate pattern recognition analysis was performed using principal component analysis (PCoA) and orthogonal projection to latent structures discriminant analysis (OPLS-DA) using the R platform to reduce the dimensionality of the multidimensional dataset and understand global metabolic changes between the fermented and unfermented feed samples. The fitting validity and predictive ability of the selected OPLS-DA model were assessed by the parameters R2Y and Q2Y, respectively. A fold change > 1, variable importance in projection score > 1.0, and *p* < 0.05 were used as the criteria to identify differential metabolites between pairwise comparison groups. To interpret the biological significance of metabolites, metabolic pathway analyses were performed by an online analysis platform in MetaboAnalyst 5.0 (http://www.metaboanalyst.ca/; accessed on 19 August 2022). Further, Kyoto Encyclopedia of Genes and Genomes (KEGG) pathway enrichment analysis was performed to annotate metabolites by matching the exact molecular mass data (*m/z*) of the samples to those from the database.

### 2.6. Correlation Analysis

The LEfSe method was applied to select the main differentially abundant genera between different samples, and KEGG pathway enrichment analysis was performed to identify significantly altered metabolic pathways. Spearman’s rank correlation coefficient was calculated with R v3.6.3 to evaluate the relationship among metabolites and microbiota.

### 2.7. Statistical Analysis

Values represent mean ± SD (n = 3 for microbial analysis, n = 6 for chemical and metabolic analyses). Data were analyzed with SPSS 22.0 (IBM Corp, Armonk, NY, USA). Statical differences between fermented and unfermented feed samples were determined by Student’s *t*-test and one-way ANOVA. LC–MS/MS data were analyzed using Proteome Discoverer (v2.4.0.305) and the built-in Sequest HT search engine. The *p*-values of metabolomics and microbiome data were corrected using Welch’s test and the Benjamini–Hochberg false discovery rate. A *p*-value of <0.05 indicated statistically significant differences.

## 3. Results

### 3.1. Chemical Composition of Fermented Feed

Table 1 showed the chemical composition of unfermented and fermented CSW samples. In comparison with unfermented feed, CP and TCA-SP levels were significantly higher in the fermented mixed feed (*p* < 0.05); in contrast, glycinin and β-conglycinin contents (*p* < 0.05) were markedly lower in the fermented mixed feed. Moreover, fermented CSW had an approximately thirty-fold increase in lactic acid content (*p* < 0.05), and pH levels decreased to 3.4.

### 3.2. Soybean Antigenic Protein Degradation

SDS-PAGE patterns of extracted soybean proteins from unfermented and fermented feed samples are shown in Appendix A. Unfermented feed samples showed the presence of multiple protein bands in the range of 35–80 kDa. Soybean antigenic protein subunits, including the α′, α, and β subunits of β-conglycinin, and the acidic and basic subunits of glycinin were separated. Fermentation with *B. licheniformis* Y5-39, *B. subtilis* B-1, and lactic acid bacteria RSG-1 significantly affected the characteristics of proteins in mixed CSW. The α′ and α subunits of β-conglycinin and the acidic subunit of glycinin in CSW feed were almost completely degraded after SSF. In the fermented feed, the number of small peptides (<25 kDa) was higher; however, there was little effect on the β subunits of β-conglycinin. Using LC–MS/MS, the differences in proteins and peptides in CSW after fermentation were determined, as shown in Table 2.

### 3.3. Microbial Diversity Analysis of Fermented Feed

The composition of the microbial community between unfermented and fermented feed samples is shown in Figure 1. The Chao1 (species richness) and Shannon (diversity) indices were used to measure α-diversity. The Chao1 index decreased while Shannon index increased in the fermented feed after SSF by compound strains; however, no significantly different changes were present between the fermented and unfermented groups (Figure 1a). This result indicated that species richness did not show any statistically significant differences between unfermented and fermented feed samples (Figure 1b). Comparing the β-diversity of microbial communities provides insights into their composition and highlights the distance or dissimilarity between each sample. Based on the Bray–Curtis distance metric and weighted UniFrac similarity method, we found that microbial β-diversity was different at different time points as the structures of microbial communities were separated into two distinct clusters (Figure 1c). The predominant bacteria changed from *uncultured_bacterium_f_Enterobacteriaceae* (5.81%), *Pantoea* (2.39%), and *uncultured_bacterium_f_Muribaculaceae* (1.25%) in the unfermented feed to *Pediococcus* (39.51%), *Enterococcus* (34.95%), and *Lactobacillus* (2.12%) in the fermented feed (Figure 1e). *Pediococcus* (*p* = 0.000), *Lactobacillus* (*p* = 0.0005), *Enterococcus* (*p* = 0.0055), and *Blautia* (*p* = 0.0253) were significantly more abundant in the fermented feed; however, the abundance of *uncultured_bacterium_f_Enterobacteriaceae* (*p* = 0.0018), *Enterorhabdus* (*p* = 0.0023), and *uncultured_bacterium_f_Muribaculaceae* (*p* = 0.0104) was lower in the fermented feed (Figure 1e). LEfSe analyses indicated that the unfermented and fermented feed samples contained significantly different bacteria, from the order to the genus level (LDA threshold > 4.0) (Figure 1d). The abundance of *Lactobacillales*, *Enterococcus*, and *Pedicoccus* was significantly higher in the fermented feed, whereas *Pantoea*, *uncultured_bacterium_o_Chloroplast*, *Protebacteria*, and *Cyanobacteria* were predominantly present in the unfermented feed (Figure 1d).

The functional potential of microbial communities was predicted in the fermented and unfermented feed samples using PICRUSt2. KEGG pathway enrichment analyses revealed 30 significant pathways in the fermented feed. Moreover, five significantly different KEGG pathways were present in the fermented feed samples, namely peptidoglycan biosynthesis, carbon metabolism, pyruvate metabolism, pentose phosphate pathway, and glycolysis/gluconeogenesis; of these, peptidoglycan biosynthesis was the most significant pathway (Figure 2). In the unfermented feed, the most significant pathways were photosynthesis; oxidative phosphorylation; porphyrin and chlorophyl metabolism; alanine, aspartate and glutamate metabolism; and methane metabolism.

### 3.4. Cluster Analysis of Metabolomics Data

To investigate metabolic changes in the fermented feed, fermented and unfermented feed samples were analyzed by LC–MS/MS-based untargeted metabolomics, which led to the identification of 699 metabolites (both positive and negative ion modes). We then performed PCA to assess variations between samples and to identify trends of distribution as well as discrete points. PCA revealed significant differences between the fermented and unfermented feed samples (Figure 3a,c). In particular, our OPLS-DA model revealed metabolic profile differences between the fermented and unfermented feed samples (Figure 3b,d), suggesting that fermentation leads to significant biochemical changes in the fermented feed. Moreover, the R2Y value of the OPLS-DA models was 1.000 and the Q2Y value was 0.995 in the positive ionization modes (Figure 3b) and the R2Y value was 0.997 and Q2Y value were 0.955 in the negative ionization modes (Figure 3d), highlighting that the models showed high reliability and good predictive capabilities. These results indicated that the model was robust and could be used to explore changes in metabolite levels over time in the fermented CSW feed.

### 3.5. Identification and Analysis of Differential Metabolites

Based on variable importance in the projection score > 1.0 and fold change >1 or <0.5 with *p* < 0.05, we identified 699 significantly different metabolites between the fermented and unfermented feed samples, of which, 299 were down- and 400 were upregulated in the fermented feed samples (Figure 4). These metabolites were mainly involved in arginine and proline metabolism, cysteine and methionine metabolism, phenylalanine metabolism, tryptophan metabolism, and protein digestion and absorption; the top 20 KEGG pathways were retained (Figure 5). Partly important metabolites were showed as major compounds contributing to the discrimination of the fermented samples after different fermentation periods, including amino acids and their derivatives, organic acids, and sugars and their derivatives (Table 3). The top 10 up- and downregulated metabolites are shown in Figure 6. Some amino acids (lysyl–valine, phenylalanyl–arginine, and Ile–Ala–Arg), organic acids (pyridoxamine-5′-phosphate and orotidylic acid), and immune-response-related metabolites (12-oxo-LTB4) were significantly upregulated in the fermented feed.

### 3.6. Correlation among Significantly Different Microbiota, Nutritional Indices, and Metabolites

To further study the effects of changes in bacteria, nutritional indices, and metabolites, we performed correlation analyses. *Pediococcus*, *Lactobacillus*, and *Enterococcus*, the abundance of which was higher in the fermented feed, showed a positive correlation with nutritional indices. Furthermore, *Pediococcus*, *Lactobacillus*, and *Enterococcus* were positively correlated with lactic acid, TCA-SP, and CP and negatively correlated with pH, β-conglycinin, and glycinin (Figure 7). Correlation analyses were also performed with the relative abundance of metabolites in the fermented and unfermented feed (Figure 7). In general, *Pediococcus* was positively correlated with 4-amino-5-hydroxymethyl-2-methylpyrimidine and malvidin 3-O-(acetylglucoside), which contribute to the nutritional value and immune function of metabolites. Further, *Enterococcus* was positively correlated with procaterol, orotidylic acid, and 4-amino-5-hydroxymethyl-2-methylpyrimidine, and *Lactobacillus* was positively correlated with lactic acid, lysyl–valine, malvidin 3-O-(acetlyglucoside), 12-oxo-LTB4, 4-amino-5-hydroxymethyl-2-methylpyrimidine, and procaterol orotidylic acid. The abundance of *uncultured_bacterium_f_Enterobacteriaceae* and *uncultured_bacterium_f_Muribaculaceae* was significantly decreased in the fermented feed, while they were the dominant bacteria in the unfermented feed. A positive correlation was observed between *uncultured_bacterium_f_Enterobacteriaceae* and daidzin, phenyl acetate, N3-fumaramoyl-L-2,3-diaminopropanoate, and PS (14:0/19:0). Collectively, these data indicated that depending on the microbial strains, metabolites affect the quality of fermented feed, possibly due to ecological changes in the microbiota.

## 4. Discussion

### 4.1. Chemical Composition of Fermented Feed and Antigen Protein Degradation

Considering the potential health benefits of microbial fermented feed for animals, its development and application have gained attention in recent times. Fermented soybean and corn–SBM-based foods are popular in animal production as they markedly contribute to animal growth. Fermentation is widely known to reduce antinutritional factor levels, produce diverse enzymes, and improve the nutritional status of substrates [31,32,33,34]. These results were consistent with changes in nutrient levels; β-conglycinin and the glycinin content showed a sharp decrease, indicative of the degradation of large-molecular-weight antigenic proteins into small peptides that contribute to higher levels of CP and TCA-SP in the fermented feed. Fermented CSW showed increased amounts of CP and TCA-SP in this present study, consistent with previous research on FSBM [35], TCA-SP consists of small peptides with 2–20 residues, which significantly increased, consistent with the degradation of β-conglycinin and glycinin in fermented soybeans. In our research, SDS-PAGE and LC–MS/MS showed significant degradation in β-conglycinin and glycinin. There were peptides and unique peptides (protein sequence coverage > 20%) largely from glycinin and β-conglycinin subunits, indicating that many amino acid residues were hidden to increase the possibility of masking epitopes, reducing fermented feed allergenicity [36]. Previous studies have shown that TCA-SP increased 4.6-fold, and β-conglycinin and glycinin decreased after 24 h of fermentation by *Bacillus subtilis* BS12, with high protease activity [37]. Glycinin and β-conglycinin are the feed allergens that are believed to be responsible for provoking transient hypersensitivity and inflammation in younger animals [38]. The low content of glycinin and β-conglycinin in fermented feed contributed to a higher feed intake of piglets [3], which resulted in better average daily weight gain. High lactic acid levels related to pH value decreased in the fermentation process.

### 4.2. Microbial Diversity and Data Cluster Analysis of Fermented Feed

We adopted a multi-omics approach to evaluate the association between microflora and metabolite production during the SSF of corn–SBM feed using compound strains. *Pediococcus*, *Enterococcus*, and *Lactobacillus* were predominant in the fermented corn–SBM feed, and amino acid metabolism was the main pathway; however, other pathways, including arginine and proline metabolism, cysteine and methionine metabolism, phenylalanine metabolism, and tryptophan metabolism were also significantly enriched during fermentation. Correlation analysis suggested a strong association between *Pediococcus*, *Enterococcus*, and *Lactobacillus* and amino acid metabolism, indicating that fermentation contributed to the nutritional value of the substrates.

The fermented feed samples showed no significant changes in α-diversity. In contrast, β-diversity was altered in the fermented feed samples, indicating that the SSF of corn–SBM feed was associated with changes in the microbial community structure and composition. Although SSF induced significant changes in the bacterial population, negligible changes were observed in the overall level of bacterial diversity. McGarvey et al. (2013) [39] reported similar results when they observed significant changes in the bacterial population structure during the ensiling of alfalfa and it’s subsequent exposure to air. Such findings suggest that significant changes in bacterial populations cannot lead to altered diversity indices, but changes in the level of some taxonomic groups can be offset by opposite changes in other groups.

In this study, microbial analyses revealed that SSF led to changes in microbial communities, which seemed to be the core reason for alterations in metabolite profiles [12]. A previous study found *Bacillus* and *Enterococcus* to be the predominant bacteria in the anaerobic stage [6]. We observed that the predominant bacteria changed from *uncultured_bacterium_f_Enterobacteriaceae*, *Pantoea* and *uncultured_bacterium_f_Muribaculaceae* in the unfermented feed to *Pediococcus*, *Enterococcus* and *Lactobacillus* in the fermented feed. *Enterococcus* and *Lactobacillus* are members of the *Firmicutes* phylum, and *Pediococcus* is a member of the *Lactobacillus* genus, which is known to contribute to quickly improving lactic acid quantities at the early stage of silage and rapidly reduce pH [40]. *Firmicutes* evidently not only secrete hydrolases (protease, lipase, and amylase) but also produce bacteriocins [41,42], which might contribute to an increase in metabolic end-product levels [43].

### 4.3. Identification and Analysis of Differential Metabolites

Our data indicated that amino acid, carbohydrate, and lipid metabolism were predominant during the fermentation process. Further, fermentation increased levels of organic acids, polysaccharides, dipeptides, and tripeptides, as well as decreasing the sugars, lipids, and carbohydrates; this finding corresponds to the results of a previous study on fermented corn and soybean [6].

The nutritional value of proteins is highly correlated with their amino acid composition; this is particularly applicable to essential amino acids in the diet of animals. Fermentation can improve the levels of amino compounds, including phenylalanyl–asparagine, serine and isoleucyl–histidine, as previously reported by Wang et al. (2022). Lysine, phenylalanine, isoleucine, and valine are essential amino acids in animal feed. Lysyl–valine, Ile–Ala–Arg, phenylalanyl–isoleucine, and glutamyl–valine levels were due to the incomplete breakdown products of protein digestion and catabolism, and these levels showed a significant increase in the fermented feed in this study; this result was consistent with that of a previous study [6]. Moreover, d-Serine is a unique endogenous substance that can improve the secretion of growth hormone; it appears to increase the N-methyl-d-aspartate-type glutamate receptor expression and increase the plasma GH levels [44]. Amino acid composition is widely known to contribute to the characteristic taste and flavor of fermented products [45], for example, serine affects sweetness, and phenylalanine, tyrosine, isoleucine, and valine contribute to a bitter taste [33]. Glutamic acid and its salts are the principal agents that impart a delicious flavor to fermented soybean products [46]. Levels of aromatic amino acids (phenylalanine, tryptophan, and tyrosine), which are vital components of protein synthesis and are found upstream of some growth hormones and secondary metabolites, were significantly increased in the fermented feed [47]. In this study, the increase in these amino acid levels in the fermented feed was probably related to methionine released from soybean proteins [48]. Furthermore, lysine degradation pathways, protein digestion and absorption pathways, and aromatic compound degradation pathways were significantly enriched after fermentation, which seems to be related to the synergistic activity of hydrolases of *B. subtilis* or fungi in the fermented feed [49,50]. However, we found that arginine and proline metabolism, cysteine and methionine metabolism, phenylalanine metabolism, and tryptophan metabolism were predominant, which led to an increase in amino acid levels in the fermented CSW feed. Phenylamine metabolism is closely related to the flavor substances produced in the fermentation procession. Phenylalanine can be converted into cinnamic acid by phenylalanine ammonia lyase, and cinnamate 4-hydroxylase continues to convert cinnamic acid into various aromaticity organic acids [51].

The level of isocitric acid, an important intermediate compound of the Krebs cycle, was significantly decreased upon fermentation. In addition, intermediate products (linolenic acid) in the aerobic decomposition of lipids were significantly increased, indicating that anaerobic fermentation plays a decisive role during the later phase of fermentation [52]. Organic acid (acetic acid) and fatty acid (propionic acid, hexanoic acid, nonanoic acid, isovaleric acid, and 3-hydroxyisobutyric acid) levels showed an improvement and were correlated with an increase in lactic acid levels and a decrease in pH in the fermented CSW feed. These products mainly serve as antibacterial elements in the fermented feed [53] as they inhibit the growth of pathogenic bacteria and reduce diarrhea in livestock, including piglets [54]. Daidzin and 6-O-acetyldaidzin, which are natural organic compounds in soybean and soybean-based products, are poorly absorbed with a low bioavailability in the gut [55]. A decrease in daidzin and 6-O-acetyldaidzin levels indicates that fermentation improves β-glucosidase activity, increasing isoflavone bioavailability and facilitating protein digestion [56]. However, levels of 4-hydroxycinnamic acid, which plays a role in the preventive effects against free radicals causing lesions and injuries, showed a significantly increasing trend in a previous study [57]. Notably, a sharp increase in the levels of 12-oxo-LTB4, PGG2, and 15-HPETE were observed in the fermented feed; they clear oxygen free radicals or inactivate oxygen free radicals the through cyclooxygenase and lipoxidase pathways of peroxidase, exerting anti-inflammatory activity [58]. Phenylacetaldehyde is a typical fragrant compound. Furthermore, 4-Amino-5-hydroxymethyl-2-methylpyrimidine is the precursor for thiamin synthesis; in a previous study, thiamin was synthesized by 4-amino-5-hydroxymethyl-2-methylpyrimidine in *B. subtilis* [59]. Malvidin 3-O-(acetylglucoside) is an anthocyanin, a type of flavonoid, and it is currently recognized as an important nutrient and natural antioxidant with high efficiency in scavenging free radicals [60]. Phenylacetaldehyde levels showed a considerable increase, which verifies that fermentation contributes to improving the nutritional value of CSW feed. Furthermore, fermentation caused a significant reduction in the content of stachyose, raffinose, glucosan, and maltopentaose; however, there was an increase in the content of β-D-galactose and uridine diphosphate glucose. This result is partly consistent with the fermentation profile of *Saccharomyces cerevisiae* reported by Sun et al. (2017) [50]. During SSF, sugars supply energy for rapid cell growth and division and bacterial cells rapidly proliferate. Stachyose changes the microbial population and bacterial enzyme activity in the intestine [61], which reportedly leads to diarrhea in piglets and also a poor growth rate [62]. We found that α-oligosaccharide (stachyose and raffinose) levels were lower in the fermented feed; α-oligosaccharides may be used by fermented strains to produce short-chain fatty acids (acetic acid, propionic acid, and butyric acid), which provide nutrition to intestinal cells and promote the growth and reproduction of probiotics in the intestine.

### 4.4. Correlation among Significantly Different Microbiota, Nutritional Indices, and Metabolites

The quality and metabolite composition of fermented feed are affected by the activities of microorganisms. *B. subtilis*, as the predominant fermentative bacteria, contributes to feed flavor and degrades large-molecular-weight components into smaller ones in fermented feed [63]. Zhang et al. (2018) evaluated the effects of SBM fermented with *B. subtilis* BS12 on the growth performance and small intestinal immune status of piglets and found that 92.36% of glycinin and 88.44% of β-conglycinin were eliminated [3]. *B. licheniformis* belongs to the *Firmicutes* phylum and showed strong protease and amylase activities; the significant degradation of glycinin and β-conglycinin was also demonstrated in the fermented feed [47]. As per previous studies, the α and α′ subunits of β-conglycinin are the preferred substrates for the majority of lactic acid bacteria, and they increase the level of total free amino acids and principally hydrolyze essential and flavor precursor amino acids [48]. We observed that glycinin and β-conglycinin levels in the fermented feed showed a significant decrease and TCA-SP levels showed a sharp increase. Protein hydrolysates are physiologically better than intact proteins because their intestinal absorption is evidently more effective [64].

*Enterococcus* and *Lactobacillus* play key roles in immune function by downregulating the expression of proinflammatory genes and upregulating that of anti-inflammatory genes in the feed of beef steers fed supplemental fermentation products of *S. cerevisiae*, *E. lactis*, *B. licheniformis*, and *B. subtilis* [65]. Moreover, *Enterococcus* and *Lactobacillus* reportedly improve proteolysis, increase the synthesis of aromatic compounds, and degrade carbohydrates, enhancing the taste and flavor of fermented feed [6]. Lactic acid levels are associated with a higher relative abundance of *Pediococcus* [62], which validates that SSF increases lactic acid levels in fermented feed. In fermented feed, lysyl–valine and lysyl–proline levels were found to be positively correlated with *Enterococcus* and *Lactobacillus* abundance. These results indicate that protein degradation and amino acid synthesis are mainly involved in fermentation by compound strains, with *Pediococcus*, *Enterococcus*, and *Lactobacillus* playing a key role in CSW feed fermentation.

## 5. Conclusions

Solid-state fermentation with *Bacillus licheniformis* Y5-39, *Bacillus subtilis* B-1, and lactic acid bacteria RSG-1 reduced soybean antigenic proteins (glycinin and β-conglycinin), daidzin, and 6-O-acetyldaidzin in CSW mixed feed and increased the TCA-SP and CP content. Furthermore, *Pediococcus*, *Enterococcus,* and *Lactobacillus* were determined as the functional core microorganisms for amino acid metabolism and lactic acid production in the fermented feed. Furthermore, arginine and proline metabolism, cysteine and methionine metabolism, phenylalanine metabolism, and tryptophan metabolism were significantly enriched in the fermented feed. These results are pivotal to obtain high-quality and safe fermented products, as well as to screen suitable fermentation agents.

## Figures and Tables

**Figure 1 microorganisms-11-01319-f001:**
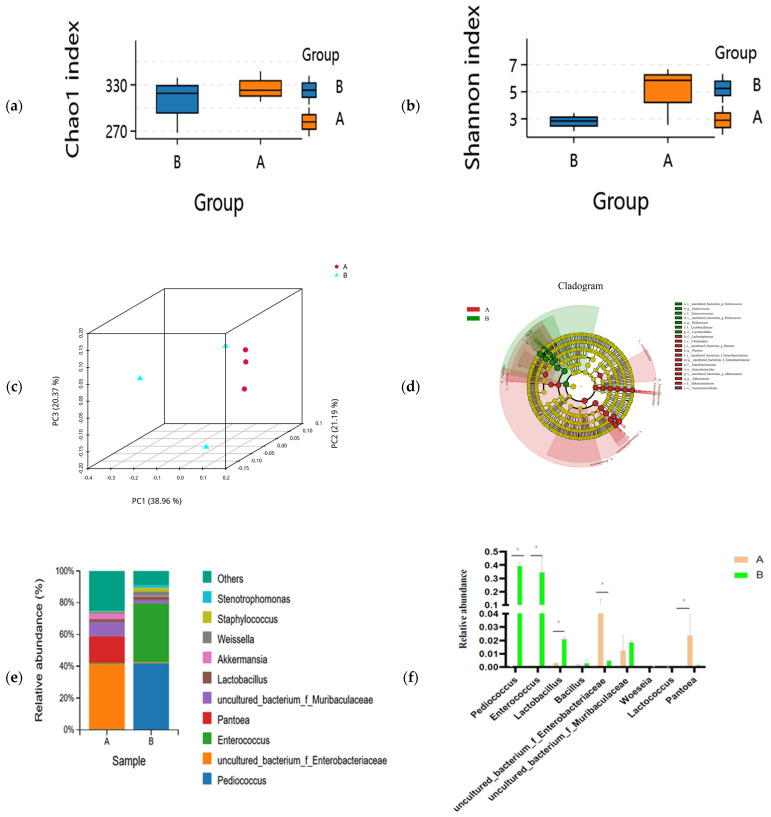
Differences in microbial community in unfermented (A) and fermented (B) feed samples. (**a**) Chao1 and (**b**) Simpson indices for unfermented (A) and fermented (B) feed samples. (**c**) Principal coordinate analysis profile of bacterial diversity in unfermented (A) and fermented (B) feed samples. (**d**) Cladogram plot of significant genera based on LEfSe analysis (LDA threshold > 4.0). (**e**) Genus-level composition of microbial community in unfermented (A) and fermented (B) feed samples. (**f**) Relative abundance of key genus in unfermented (A) and fermented (B) feed samples, graph with “*” are different (*p* < 0.05).

**Figure 2 microorganisms-11-01319-f002:**
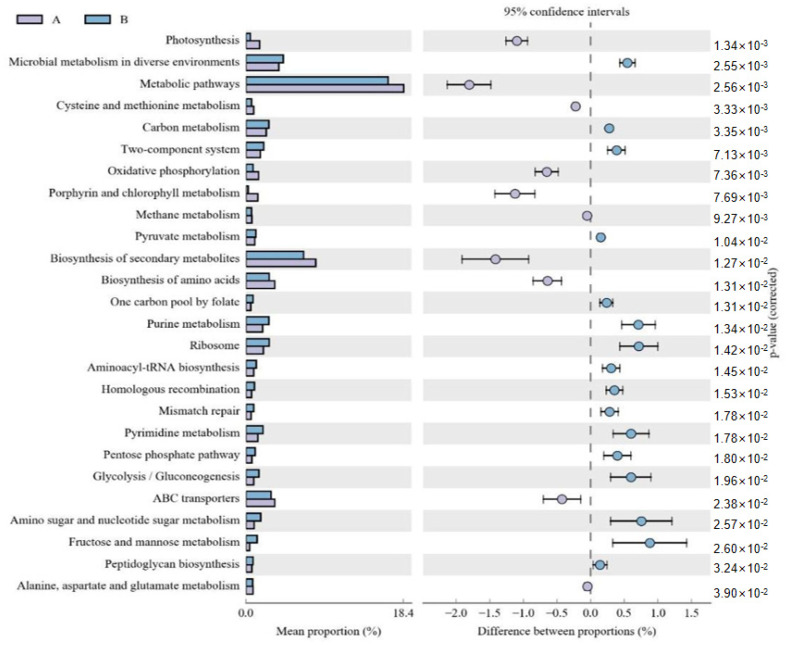
Comparison of functional pathways predicted by PICRUSt2 in unfermented (A) and fermented feed (B) samples. The extended error bar method in STAMP showing relative difference pathway in comparisons involving one hypervariable dataset and other datasets using Welch’s *t*-test (two-sided).

**Figure 3 microorganisms-11-01319-f003:**
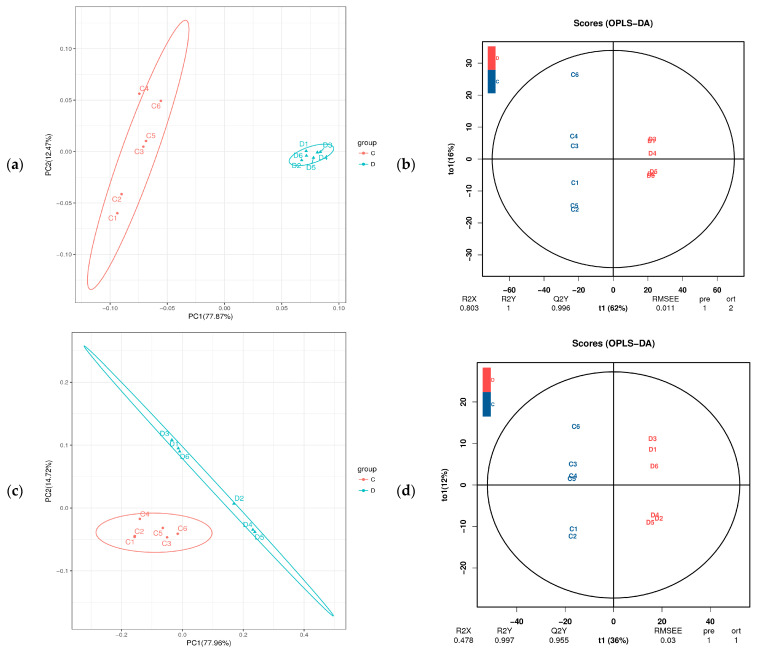
PCoA and OPLS-DA plot based on fermented feed metabolites. C: unfermented feed; D: fermented feed (**a**) PCoA and (**b**) OPLS-DA plot of metabolites in the positive ionization mode between unfermented and fermented feed samples. (**c**) PCoA and (**d**) OPLS-DA plot of metabolites in the negative ionization mode between unfermented and fermented feed samples.

**Figure 4 microorganisms-11-01319-f004:**
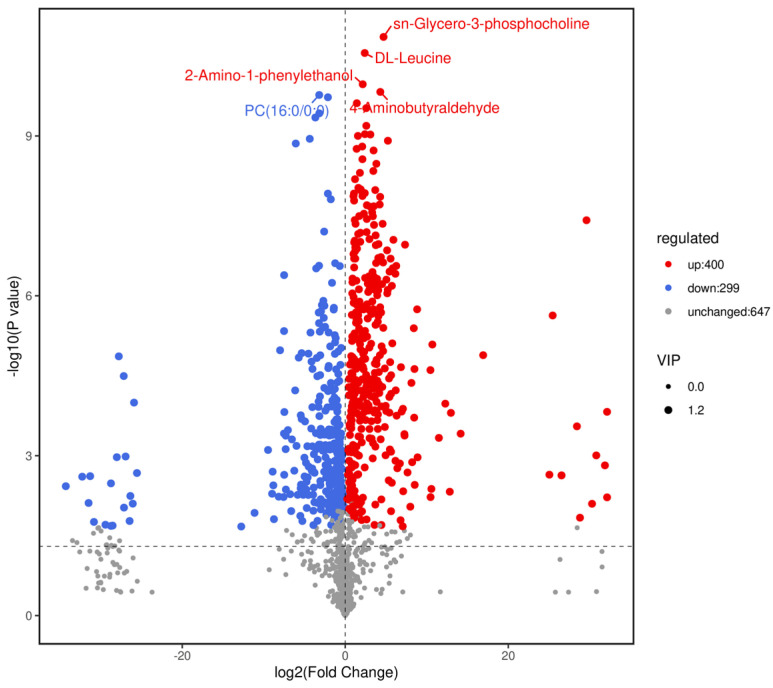
Volcano plot of differential metabolites (positive + negative ions). Fold changes (log_2_) are plotted on the abscissa and *p*-values (−log10) on the ordinate. Data points corresponding to non-differential metabolites (NoDiff), upregulated metabolites (UP), and downregulated metabolites (DW) are shown as grey, red, and blue circles, respectively. Circle size represents variable importance in projection score for corresponding metabolites.

**Figure 5 microorganisms-11-01319-f005:**
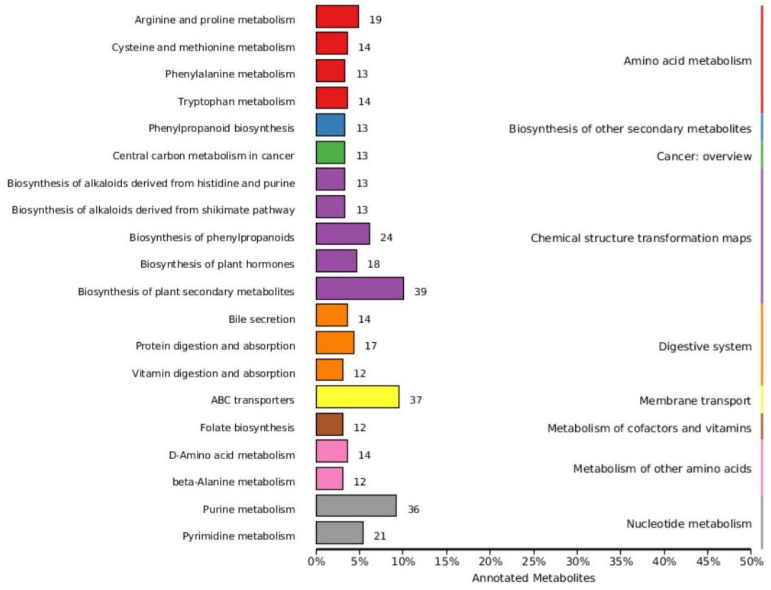
Classification map of KEGG differential metabolic pathways.

**Figure 6 microorganisms-11-01319-f006:**
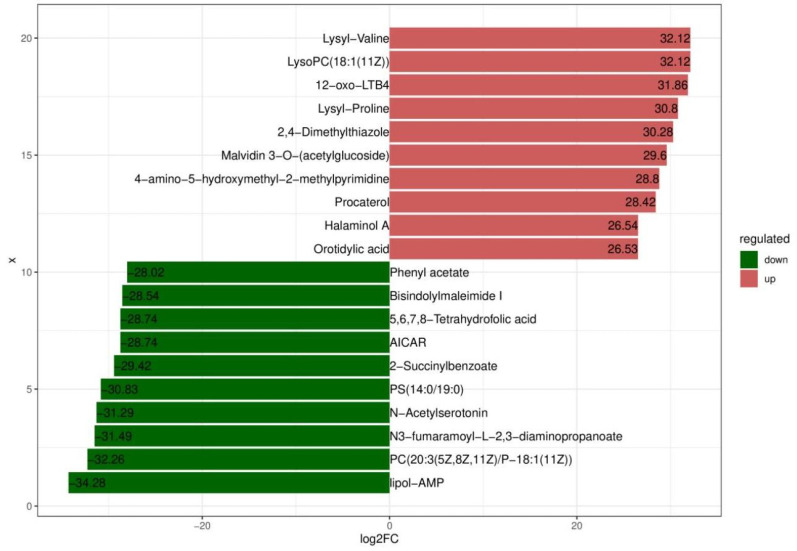
Top 10 up- and downregulated metabolites.

**Figure 7 microorganisms-11-01319-f007:**
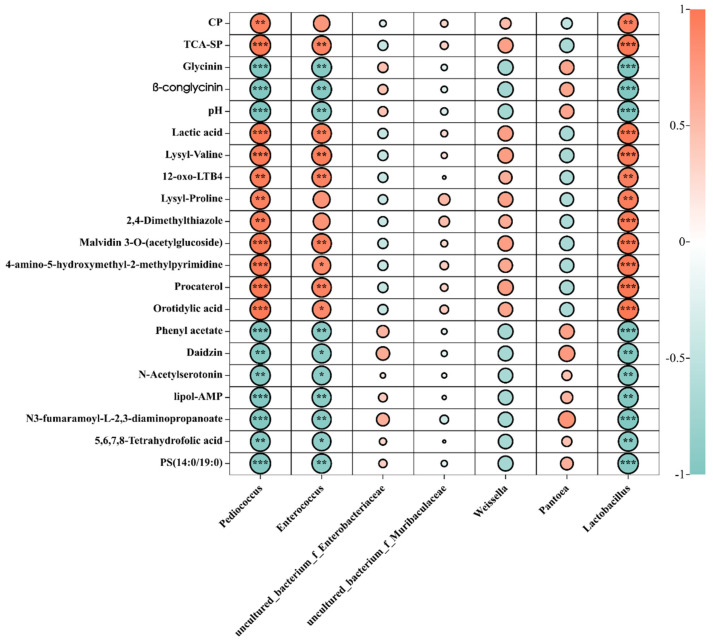
Relationships among the dominant microbiota and the first 20 metabolites in unfermented and fermented feed samples. Graph with “*” was correlation (*p* < 0.05); “**” was significantly correlation 0.01 < *p* < 0.05; “***” was extremely significant correlation (*p* < 0.01).

**Table 1 microorganisms-11-01319-t001:** Changes in the content of various compound classes in unfermented and fermented feed samples.

Items	Unfermented Feed (0 Days)	Fermented Feed (14 Days)
CP %	26.70 ± 1.29 ^A^	33.24 ± 1.82 ^B^
TCA-SP %	1.74 ± 0.40 ^A^	7.32 ± 0.79 ^B^
TCA-SP/CP	6.70 ± 1.80 ^A^	21.99 ± 1.25 ^B^
Glycinin	36.26 ± 1.38 ^B^	10.57 ± 1.27 ^A^
β-conglycinin	39.27 ± 0.94 ^B^	13.12 ± 1.54 ^A^
pH	5.99 ± 1.08 ^B^	3.40 ± 0.95 ^A^
Lactic acid g/kg	0.245 ± 0.02 ^A^	30.418 ± 0.12 ^B^

Data in the same row with different uppercases are different (*p* < 0.05). CP = crude protein; TCA-SP = trichloroacetic acid soluble protein.

**Table 2 microorganisms-11-01319-t002:** Proteins and peptides of extracted proteins in fermented feed.

Accession No.	Master Protein	Peptides	Unique Peptide	Sequence Coverage (%)	pI	Molecular Weight (kDa)
P04776	Glycinin G1	22	18	58	6.23	55.7
P02858	Glycinin G4	25	19	53	5.29	63.8
P04405	Glycinin G2	22	16	63	5.58	54.4
F7J077	β-conglycinin β-subunit 2	25	2	57	6.24	50.4
P25974	β-conglycinin β-subunit 1	25	2	57	6.24	50.4
P11827	β-conglycinin α′ subunit	23	17	36	5.71	72.2
P0DO16	β-conglycinin α subunit 1	20	12	36	5.17	70.3
P04347	Glycinin G5	19	13	51	5.9	57.9
P11828	Glycinin G3	13	8	31	5.97	54.2
P01070	Trypsin inhibitor A	9	4	38	5.11	24.0
Q04672	Sucrose-binding protein	10	10	21	6.87	60.5
P25272	Kunitz-type trypsin inhibitor KTI1	6	5	31	5.12	22.5
P05046	Lectin	6	6	32	6.05	30.9
P13867	α-Amylase/trypsin inhibitor	6	2	37	7.78	22.1
P01085	α-Amylase inhibitor	2	2	22	7.05	13.3

**Table 3 microorganisms-11-01319-t003:** The comparison of important metabolites between unfermented and fermented feed samples.

	Metabolites	Log_2_-Fold Change (Unfermented vs. Fermented Feed)	*P*	Variable Importance in Projection	Regulated	KEGG
Amino acid and its derivatives	Lysyl–Valine	32.1225	0.006042	1.139729	up	NA
	Lysyl–Proline	30.79547	0.000987	1.253102	up	NA
	L-Lysine	22.01479	8.87 × 10^−6^	1.254099	up	C00047
	Serinyl–Glutamine	17.55396	4.54 × 10^−5^	1.239443	up	NA
	Ile–Ala–Arg	16.92034	1.30 × 10^−5^	1.359839	up	NA
	Threoninyl–Arginine	13.2527	7.34 × 10^−8^	1.270448	up	NA
	Histidinyl–Tryptophan	11.56049	4.68 × 10^−8^	1.271329	up	NA
	Phosphatidylinositol 4,5-bisphosphate	11.48888	0.000442	1.17491	up	NA
	Arachidonic Acid	12.82632	0.004739	1.155216	up	NA
	Phenylalanyl–Arginine	7.816741	8.64 × 10^−8^	1.269362	up	NA
	Glutamyl–Valine	6.699018	0.001401	1.239139	up	NA
	Acetyllysine	6.410332	5.82 × 10^−6^	1.257408	up	C12989
	Threoninyl–Lysine	6.298735	0.000177	1.315232	up	NA
	Phenylalanyl–Isoleucine	6.25610123	2.75 × 10^−7^	1.375190642	up	NA
	Glycine	3.29772931	2.72 × 10^−8^	1.363462454	up	C00037
	Prolyl–Alanine	3.634916	1.87 × 10^−5^	1.35041	up	NA
	Valyl–Phenylalanine	2.585049	6.45 × 10^−10^	1.377709	up	NA
	d-Serine	3.78452377	2.43 × 10^−7^	1.374931883	up	C00955
	5-Hydroxy-L-tryptophan	0.506922	0.0009	1.11541	down	C00643
	Phenylacetaldehyde	5.413254	7.88 × 10^−5^	1.334544	up	C00601
	12-oxo-LTB4	31.85966	0.001519	1.230829	up	C05949
Organic acids	Pyridoxamine 5′-phosphate	25.04453	0.002271	1.212029	up	C00647
	Orotidylic acid	26.53149	0.002344	1.209657	up	C01103
	Adenosine 2′,3′-cyclic phosphate	−25.5498	0.002113	1.214595	down	C02353
	5-Amino-6-(5′-phosphoribitylamino)uracil	−26.0651	0.007939	1.111522	down	NA
	PC(14:0/20:2(11Z,14Z))	14.15887894	0.000385945	1.292795098	up	C00157
	Arachidonic Acid (peroxide free)	12.82631899	0.004739497	1.155215501	up	NA
	Propionic acid	10.48348	3.21 × 10^−8^	1.273026	up	C00163
	Linoleic acid	7.218953	0.000142	1.217527	up	NA
	Isovaleric acid	7.332989	0.000122	1.213306	up	C08262
	α-D-Glucose 1,6-bisphosphate	5.720319	6.49 × 10^−7^	1.373191	up	NA
	2-acylglycerophosphocholine	5.713364	0.003222	1.180395	up	NA
	Pentanoic acid	3.168922	1.85 × 10^−6^	1.224405	up	C00803
	4-Hydroxycinnamic acid	3.160283	2.99 × 10^−6^	1.248225	up	C00811
	Glutaconic acid	1.201342	6.48 × 10^−9^	1.260067	up	C02214
	α-Ketoisovaleric acid	0.864318	9.00 × 10^−7^	1.236833	up	C00141
	Myristic acid	0.526749	0.000262	1.158736	up	C06424
	3-(indol-3-yl)pyruvic acid	−0.92557	0.003171	1.087165	down	C00331
	Nonanoic acid	−3.05773	0.000408	1.153522	down	C01601
	Syringic acid	−2.02246	0.001712	1.129497	down	C10833
	2-Isopropylmalic acid	−1.02382	0.005286	1.052666	down	C02504
	Isocitric acid	−1.5716	0.000408	1.166359	down	C04617
	Gamma-Aminobutyric acid	−1.41422	1.81 × 10^−6^	1.239942	down	C00334
	Ferulic acid	−0.62699	0.000767	1.133006	down	C01494
	15(S)-HPETE	0.695366	0.004925	1.034271	up	C05966
Sugar and its derivatives	Maltotriose	0.293095	1.54 × 10^−8^	1.27159	down	C00420
	1-Deoxy-D-xylulose	−26.9445	0.001035	1.254408	down	C06257
	Stachyose	0.236374	2.27 × 10^−5^	1.247808	down	C01613
	Malvidin 3-O-glucoside	0.03646	0.004313	1.074059	down	C12140
	Raffinose	0.48141	0.001225	1.138124	down	C00492
	β-D-Galactose	2.032622	0.000161	1.162937	up	C00962
	Glucosan	0.278913	8.89 × 10^−6^	1.251205	down	NA
	Maltopentaose	0.170101	1.54 × 10^−6^	1.264761	down	NA
	Apigenin 7-glucoside	0.024617	0.000204	1.210975	down	C04608
	Uridine diphosphate glucose	14.08391	3.33 × 10^−9^	1.262763	up	C00029
	Cyclotene	0.467395	0.003040945	1.096896721	down	NA

## Data Availability

Raw data for the figures are available upon reasonable request from the corresponding author.

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
