# Peer review of "Integrated Microbiota and Metabolome Analysis to Assess the Effects of the Solid-State Fermentation of Corn–Soybean Meal Feed Using Compound Strains"

_microorganisms, 2023, doi:10.3390/microorganisms11051319_

Round 1

Reviewer 1 Report

Solid state fermentation is a bioprocess where microbial organism undertakes fermentation of substrate matrix in absence of free-flowing water. Although abundant water is absent in solid state fermentation the substrate must have enough water to sustain growth of microbes. Solid state fermentation is becoming more important because of bioactive compound or secondary metabolites produced in the process. Solid sate fermentation has been used in reduction of non-starch polysaccharides and α-galactosides of soybean meal. The authors therefore inoculated corn-soybean meal feed with some Bacillus strains and the metabolomic profiling were applied to investigate changes in microflora and metabolites. The work is interesting and provides novel insights on dynamic changes in metabolism occured during solid-state fermentation of corn-soybean meal feed. The results also have the potential for practical use in fermentation production. For the most part, this work is based on sound designs and experiments.

Manuscript can be further improved taking following points into consideration,

1. 2.1 the media for cultivation of the Bacillus strains should be described.

2. Phenylalanine metabolism is the most important pathway in the fermentation process according to previous researches. However, it was not extensively discussed in the result section of this manuscript.

3. Why did the authors choose the mixed ratio of 1:4:1 for the three inoculated microorganisms? How did the authors consider the influence of the ratio on the fermentation result?

4.     The caption of Table 3 should be corrected, because it described the comparison of the results of two sample sets, not the specific amount of metabolites. 

Moderate editing of English language is needed.

Reviewer 2 Report

This manuscript investigates the microbiota and metabolome of solid-state fermentation of corn-soybean meal with compound strains, with the aim of elucidating the underlying mechanisms by which composite microbial fermentation influences the unique quality of corn-SBM feed. It is a topic of interest to the researchers in the related areas but needs moderate revisions. The detailed comments are as follows:

1. Introduction: There are a lot of papers about corn-SBM feed with compound strains, the innovation point of this manuscript should be highlighted.

2. Material and Methods: Third generation sequencing technology has been increasingly utilized to analyze microbial community. Please explain the reason for the selection of NGS sequencing instead of third generation sequencing in this manuscript.

3. Results: 3.1 “CP and TCA-SP levels were significantly higher in fermented mixed feed”, Please explain the reason.

4. Results: The title of 3.2 is material instead of result, please modify it.

5. Results: 3.2, please explain the reason for using metabolomics instead of proteomics to check the differences of proteins and peptides.

6. It is suggested that Figure 1 could be placed into supplementary material.   

7. Chemical Composition is lacking in discussion.

8. Discussion: Some writing problems, for example, abuout in the nineth line, please check the whole manuscript.

9. The conclusion needs to be further condensed.

 Minor editing of English language required.

Reviewer 3 Report

This manuscript described the effect of fermentation on bacterial community and metabolites of a mixture of corn, soybean and wheat bran, which is interesting in fermented feed filed. However, the authos must deal with the comments or suggestions before further considering acceptance.

There are a few low-levle mistakes. To begin with, line number was not used in the whole manuscript, which makes the reviewer torturous and confusing. In the following revsion process, the authors must add it. Second, the p value should be lowercase and italic in the text. Third, corn, soybean and wheat bran were used in this work, rather than "corn and soybean meal", so the authors should not use the abbreviation (corn-SBM)any more in the text, I suggest using "CSW". Last, the "Discussion" section should be divided into many parts according to the "Results" section, namely consisting of 4.1 Chemical Composition of Fermented Feed, 4.2 SDS-PAGE and LC–MS/MS, 4.3 Microbial diversity Analysis of Fermented Feed, 4.4 Cluster Analysis of Metabolomics Data, 4.5 Identification and Analysis of Differential Metabolites and 4.6 Correlation Among Significantly Different Microbiota, Nutritional Indices, and Metabolites. The comments in detail can be available as follows.

Abstract

L2, in fermented mixture of corn, soybean and wheat bran remains unclear. Because, there are a lot of references on fermented feed published on line, such as fermented forages (silage)

L3, which species for "RSG-1" ? please describe it

L9, "Overall,  699", please use "English state" throughout the text

L10, since only one time point was used in this work, so where are "different fermentation times" ?

L6 in page 2, there is no need to use the italic format for "Lactic acid"; please insert one blank between "process" and "[6]", also in the rest positions

L2 in page 3, how did the authors finish the process of transferring ? describe it in detail 

L14 in page 3, only lactic acid content determination was insufficient, acetic acid and propionic acid need to be determined too.

3.3. Microbial diversity Analysis of Fermented Feed

L4, there is no need to emphasize the effect of microbial additive, because it is hard to differentiate the effect between fermentation and microbial additive

L13, do not use the the italic format for "uncultured", also in the other positions

Reference

8, "Lactobacillus plantarum" should be italic

Overall, the quality is okay

Reviewer 4 Report

The manuscript entitled Integrated Microbiota and Metabolome Analysis to Assess the Effects of Solid-State Fermentation of Corn–Soybean Meal Feed using Compound Strains was revised and improved. In my opinion, the publication in its current version can be performed.

Minor editing of English language required.

Author Response

thanks for your comments

Round 2

Reviewer 1 Report

All the problems have been addressed.

Author Response

thank you for your comments

Reviewer 2 Report

Accept in present form.

Author Response

thank you for your comments

Reviewer 3 Report

No comments

Author Response

thank you for your comments